# Expandable Interbody Cages in 1–3 Level Circumferential Lumbar Arthrodesis with 2-Year Follow up: A Retrospective Study

**DOI:** 10.3390/bioengineering12111169

**Published:** 2025-10-28

**Authors:** Fava Marco, Vommaro Francesco, Toscano Angelo, Ciani Giovanni, Parciante Antonio, Mendola Elena, Nervuti Giuliana, Maccaferri Bruna, Gasbarrini Alessandro

**Affiliations:** 1Complex Spinal Surgery Unit, IRCCS Istituto Ortopedico Rizzoli, 40136 Bologna, Italy; marco.fava@ior.it (F.M.); francesco.vommaro@ior.it (V.F.); giovanni.ciani@ior.it (C.G.); antonio.parciante@ior.it (P.A.); giuliana.nervuti@ior.it (N.G.); alessandro.gasbarrini@ior.it (G.A.); 2Ortopedia Generale, IRCCS Istituto Ortopedico Rizzoli, Via di Barbiano 1/10, 40136 Bologna, Italy; angelo.toscano@ior.it (T.A.); elena.mendola@ior.it (M.E.); 3Department of Biomedical and Neuromotor Sciences, University of Bologna, 40127 Bologna, Italy

**Keywords:** expandable cages, lumbar spine, spine fusion, interbody fusion, spinal instrumentation

## Abstract

**Introduction**: Currently, static interbody cages are the gold standard for achieving solid arthrodesis in the spine, enhancing segmental stability, obtaining neuroforaminal decompression, and improving as well as maintaining segmental lordosis. It is well known that restoring sagittal balance and segmental lordosis is crucial for long-term outcomes in lumbar spine fusion. For some cases, expandable interbody cages are emerging as an alternative to static cages. This study aims to evaluate the radiographic outcomes and complications of standard open transforaminal lumbar interbody fusion (TLIF). **Methods**: A standard open TLIF procedure using expandable cages was performed at 1 to 3 levels in 71 patients (129 levels in total), with a follow-up of two years. All patients underwent radiological assessments preoperatively, immediately postoperatively, and at one and two years postoperatively. Radiological evaluation was conducted using standing lateral X-rays. **Results**: Segmental lordosis (SL) increased significantly from the preoperative value (9.0° ± 3.6°) to 24 months postoperatively (15.4° ± 3.0°), with improvements maintained throughout the 24-month follow-up period (*p* < 0.001). Similarly, anterior disc height (ADH), posterior disc height (PDH), and foraminal height (FH) each increased significantly from preoperative to immediate postoperative measurements, and these gains were maintained over the two-year follow-up (*p* < 0.001 each). Lumbar lordosis increased significantly from the preoperative value (41.9° ± 10.5°) to the immediate postoperative period (45.7° ± 10.8°); however, this improvement decreased slightly at the one- and two-year follow-ups. No revisions were required for cage-related complications. One patient experienced a surgical site infection, and two patients had mechanical complications (screw loosening and proximal junctional kyphosis). **Conclusions**: Expandable interbody cages enable excellent restoration and maintenance of disc height and segmental lordosis in a standard open TLIF procedures at two-year. Achieving these outcomes depends on several factors, including proper preparation of the vertebral endplates, accurate cage placement and expansion, posterior facet osteotomy, and the application of posterior compression prior to final fixation. These steps are essential to fully maximize the potential of expandable cage technology.

## 1. Introduction

Chronic low back pain that does not respond to conservative treatments often requires surgical intervention for various lumbar spine conditions, such as degenerative disc disease, spondylolisthesis, spinal stenosis, disc herniation, and degenerative scoliosis. Management strategies for these conditions frequently involve interbody fusion, aiming to reestablish spinal alignment, achieve indirect decompression of the neural foramina by restoring disc height, and improve fusion rates [1,2]. For patients with spinal stenosis and instability who require surgery, lumbar fusion remains the gold standard and has been shown to alleviate symptoms of neurogenic claudication.

Multiple approaches/techniques have been described for lumbar interbody fusion (LIF), including anterior LIF, lateral LIF, oblique lateral LIF, extreme lateral LIF, posterior lumbar interbody fusion (PLIF), transforaminal lumbar interbody fusion (TLIF), and minimally invasive LIF (MIS-LIF) [3]. For many years, neutral static interbody cages were considered the gold standard implants for interbody fusion procedures. These devices facilitated spinal fusion while allowing for the restoration of disc height and foraminal height (FH), improvement of sagittal alignment, and support of spinal stability. However, the fixed height and non-lordotic configuration of static cages limit their ability to fully restore anterior column height to anatomical levels, potentially restricting postoperative pain relief. Additionally, intervertebral distraction is typically required to enable trialing and insertion of static cages, necessitating nerve root retraction and increasing the risk of dural tears and iatrogenic nerve injury. Also, in many cases, placement of a static cage requires aggressive endplate preparation and forceful impaction, which can lead to endplate damage and, consequently, cage subsidence [4].

To address these limitations, lordotic expandable cages were introduced. These devices can be inserted in a collapsed form, eliminating the need for trialing or vertebral distraction, thereby minimizing trauma to the endplates and theoretically reducing the risk of implant subsidence while achieving optimal interbody height. Also, cages with larger footprints have developed to increase the contact surface area, improve fusion rates, and preserve the relationship between the vertebral endplates [5,6,7].

Although expandable cages were developed to overcome the limitations of static cages, the literature reports conflicting findings regarding the use of expandable cages, particularly in terms of radiographic outcomes [3,4,5,6,8].

The purpose of the present study was to provide results that resolve this conflict. Thus, we documented radiographic outcomes and complications over a 2-year follow-up period, with specific attention to segmental lordosis, lumbar lordosis, disc height restoration, and postoperative cage subsidence in 71 patients who underwent one- to three-level circumferential arthrodesis using expandable interbody cages.

## 2. Methods

### 2.1. Patient Population

This was a retrospective observational study conducted on a cohort of patients who underwent single-, two-, or three-level continuous TLIF at a single institution between 2017 and 2023, using one of the three designs of an expandable interbody cage: L-Varlock (Kisco International, Saint Priest, France), Rise (Globus Medical, Audubon, PA, USA), or Uplifter (Fule Science & Technology Development (Beijing, China) and Invibio (Thornton-Cleveleys, UK)) (Figure 1).

Inclusion criteria: age over 18 years, disease at one, two or three contiguous levels who have failed conservative treatment (at minimum duration of 6 months), surgical treatment consisting of one- to three-level continuous circumferential lumbar arthrodesis with a minimum radiographic follow-up of two years.

Exclusion criteria: patients under 18 years old, trauma, tumor, prior fusion and/or surgical operation using other interbody cages than expandable cages mentioned above (Table 1).

### 2.2. Surgical Procedures

All cases were performed using conventional open TLIF procedures. Under general anesthesia, patients were placed prone on an Allen spine system bed. The surgical site was prepared and draped in the standard sterile manner, and antibiotics were administered according to hospital guidelines.

A midline incision was made, followed by subperiosteal exposure of the posterior elements extending to the transverse processes and sacral ala. Pedicle screws were inserted bilaterally using anatomical landmarks and fluoroscopic guidance. Laminectomy was performed, followed by neural decompression. After facetectomy, the disc space was accessed on the most symptomatic side.

Following gentle retraction of the neural elements, annulectomy was carried out, and discectomy was performed using pituitary rongeurs and disc shavers to thoroughly evacuate disc material. The endplates were carefully prepared using a rasp. If necessary, distraction was applied across the screws. The interbody cage, packed with appropriately prepared autologous bone graft, was then placed into the disc space and fully expanded in lordosis. Additional bone graft was packed dorsally to the cage.

Appropriately sized rods were shaped and placed. The wound bed was irrigated with saline and closed over a drain.

### 2.3. Radiographic Measures

Demographic data were collected for each patient. Radiographic parameters were evaluated at the following time points: preoperative X-ray, immediate postoperative, 12-month, and 24-month follow-up.

The radiographic parameters included lumbar lordosis (L1–S1, LL), segmental lordosis (SL), anterior and posterior disc height (DH), and foraminal height (FH) at the operative level(s). All measurements were performed using the Picture Archiving and Communication System (PACS) by a single operator (Figure 2). To assess measurement reliability, all radiographic evaluations were independently repeated by a second operator after a washout period of at least two weeks.

Anterior and posterior intervertebral disc heights were measured using standing neutral lateral radiographs. Segmental lordosis was determined by the angle formed between tangent lines drawn along the superior endplate of the superior vertebra and the inferior endplate of the inferior vertebra at each treated level. Foraminal height was measured as the vertical distance from the inferior pedicle wall of the vertebra above to the superior pedicle wall of the vertebra below (Figure 3).

Subsidence was evaluated using standing neutral lateral radiographs with parallel endplates at the index level. It was classified according to the extent of cage subsidence into the vertebral endplates as follows: Grade 0, corresponding to 0–24% collapse; Grade I, 25–49%; Grade II, 50–74%; and Grade III, 75–100% collapse of the treated level. Grades 0 and I were categorized as low-grade, while Grades II and III were considered high-grade subsidence. Subsidence was assessed on immediate postoperative radiographs, and at the 12-month and 24-month follow-ups.

### 2.4. Statistical Analysis

Descriptive statistics were calculated for all variables. Continuous data are presented as means ± standard error of the mean (SEM) or as medians with interquartile ranges (IQRs), depending on data distribution. Categorical variables are reported as frequencies and percentages.

Normality was assessed with the Shapiro–Wilk test. Comparisons between groups were performed using Student’s *t*-test, Mann–Whitney U test, or Kruskal–Wallis test, as appropriate. Repeated measures ANOVA with Bonferroni correction was subsequently performed to account for within-subject variability and to minimize type-I error inflation associated with multiple paired t-tests across time points. Effect sizes (η^2^) were also calculated to quantify the magnitude of observed differences. Statistical significance was defined as *p* < 0.05. All analyses were performed using Jamovi statistical software (version 2.6.26).

## 3. Results

### 3.1. Demographics

A total of 81 patients were initially enrolled in the study, of whom 71 were available for radiographic measurements at 24 months follow-up. Among these, 29 patients underwent open single-level circumferential arthrodesis, 26 patients underwent two-level open circumferential arthrodesis, and 16 patients underwent three-level open circumferential arthrodesis.

In the single-level group, procedures were performed at the following levels: L5-S1 (18 cases), L4-L5 (8 cases), L3-L4 (2 cases), and L2-L3 (1 case). In the two-level group, 21 cases involved levels L4-S1 and 5 cases involved L3-L5. In the three-level group, 12 cases involved levels L3-S1 and 4 cases at L2-L5.

The mean age of patients at the time of surgery was 56 years (range, 18–80 years). The cohort included 30 males and 41 females, with a mean body mass index (BMI) of 25.5 kg/m^2^ (range, 18.3–32.4 kg/m^2^) (Table 2).

### 3.2. Radiographic Results

All the results are summarized in Table 3 and Figure 4. Among the evaluated radiographic parameters, segmental lordosis (SL) increased significantly from the preoperative value (9.0° ± 3.6°) to 24 months postoperatively (15.4° ± 3.0°), and this increase was maintained throughout the 24-month follow-up period (*p* < 0.05) (Table 3). Lumbar lordosis (LL; L1-S1) also increased significantly from the preoperative mean of 41.9° ± 10.5° to 45.7° ± 10.8° immediately postoperatively; however, this improvement decreased at the 1-year (42.3° ± 11.4°) and 2-year (41.7° ± 11.6°) follow-ups. All statistical analyses (paired *t*-tests, non-parametric tests, and repeated-measures ANOVA with Bonferroni correction) yielded consistent results, confirming that the statistical significance of radiographic changes remained unchanged regardless of the test applied.

Similarly, anterior disc height (ADH), posterior disc height (PDH), and foraminal height (FH) showed statistically significant improvements from preoperative to immediate postoperative measurements, and these changes were sustained over the 24 months (*p* < 0.001 each). On average, SL improved by approximately 70% at both 1-year and 2-year follow-ups. ADH, PDH, and FH increased by approximately 81%, 150%, and 90% at 1 year, and 70%, 135%, and 80% at 2 years, respectively.

Subsidence was assessed and classified based on the extent of cage subsidence into the vertebral endplates, evaluated on immediate postoperative, 12-month, and 24-month radiographs. At the last follow-up, low-grade subsidence (Grades 0 and I) was observed in 94.4% of patients (67/71), with 53 cases classified as Grade 0 and 14 as Grade I. High-grade subsidence (Grades II and III) was observed in 5.6% of patients (4/71), with 3 cases of Grade II and 1 case of Grade III. No patient required revision surgery due to cage subsidence.

### 3.3. Complications

One patient developed a surgical site infection and required reoperation two months after the initial procedure. Two patients experienced mechanical complications: one had screw loosening and underwent implant revision 22 months after the initial surgery, while the other developed proximal junctional kyphosis (PJK) 24 months postoperatively and underwent revision with proximal extension of the implant.

## 4. Discussion

Restoring appropriate sagittal alignment and lumbar lordosis is widely recognized as a key factor in improving long-term outcomes following short-segment lumbar fusion [9,10,11]. However, there is still a lack of consensus in the literature regarding the most effective type of interbody cage—expandable versus static—for achieving and maintaining segmental lordosis, disc height, and global lumbar alignment. Moreover, studies give conflicting findings regarding the use of expandable cages, particularly in terms of radiographic outcomes [3,4,5,6]. The purpose of this study was to clarify conflicting evidence by evaluating radiographic outcomes and complications over a 2-year follow-up in patients undergoing one- to three-level circumferential arthrodesis with expandable cages. Specifically, we assessed changes in segmental lordosis, lumbar lordosis, anterior and posterior disc height, foraminal height, and the incidence of cage subsidence.

In our study, segmental lordosis demonstrated a consistent relevant increasing trend following surgery. Our data showed a mean gain of 6.3 degrees after the use of expandable cages, with this improvement remaining stable over the 2-year follow-up period without statistically significant loss. This suggests that expandable cages are effective in achieving and maintaining segmental alignment with open TLIF procedures. Several factors may contribute to this sustained correction. In degenerative spine fusion, SL is influenced by multiple variables, including the patient’s preoperative anatomy and bone quality, the surgical technique, as well as cage placement and design [11,12,13]. In particular, while static cages can improve SL when positioned more anteriorly, expandable cages allow for controlled expansion with less endplate disruption due to their smaller initial profile. This may reduce the risk of complications and may contribute to the long-term stability of segmental correction [10,14,15].

Lumbar lordosis improved significantly from the preoperative value to the immediate postoperative with a mean gain of 3.8 degrees. However, we reported that the improvement of LL in the immediate postoperative was not maintained at 2-years follow up, suggesting that expandable cages may not have a protective effect on overall lumbar lordosis. This can be partially explained by the fact that our study analyzed different levels of the lumbar spine, between L2 and S1, undergoing circumferential arthrodesis. It is known that lumbar lordosis is given for 2/3 by L4-S1 segment and for 85% by L3-S1 segment, emphasizing the greater influence of specific lumbar levels on postoperative outcomes of global lordosis [12].

Anterior disc height (ADH) showed a significant and sustained improvement following surgery. On average, we observed a postoperative gain of 5.5 mm, with minimal reduction over the 2-year follow-up period and no statistically significant loss. This suggests that expandable cages are effective in restoring anterior column height and maintaining it over time. The anterior placement of these cages and their capacity for controlled expansion may help distribute forces more evenly along the endplates, particularly toward the stronger anterior apophyseal ring, which could reduce the risk of subsidence and preserve disc height.

Posterior disc height (PDH) also demonstrated a marked and durable improvement. Our data showed a mean increase of 5.9 mm postoperatively, which remained relatively stable over the follow-up period. This restoration may be attributed to the mechanical design of expandable cages, which allows for uniform disc space distraction during deployment. Additionally, reduced endplate violation due to the cage’s low initial profile likely contributes to better preservation of posterior disc height and intervertebral stability.

Foraminal height (FH) increased significantly after surgery and remained stable over time. The mean gain was 1.3 cm, with only a slight decrease at the 2-year follow-up, which was not statistically significant. The improvement in FH is likely a direct consequence of increased disc height, which indirectly decompresses the neural foramina. Expandable cages may enhance this effect by restoring disc space more effectively, thus maintaining neural element decompression without the need for more invasive procedures.

The significant improvements in segmental lordosis (SL), anterior disc height (ADH), posterior disc height (PDH), and foraminal height (FH) observed in our study are consistent with findings from previous research on expandable cages only [16,17,18,19,20]. Notably, the degree of SL restoration in our cohort appears greater than that reported in comparable studies, whereas the postoperative improvement in lumbar lordosis (LL) tended to decline more overtime, showing a less favorable long-term trend compared to other reports in the literature. Regarding subsidence, with the radiographic classification that we used [8], we reported 5,6% of high-grade subsidence (4/71) and 94,4% of low-grade subsidence (67/71). None of these cases led to persistence of symptoms and reoperation with cage revision. This result is comparable with other studies in the literature that showed no significant increase in subsidence rate with the use of expandable cage [16,19,21]. However, direct comparisons remain challenging due to the heterogeneity of patient populations, surgical techniques, and cage designs across studies [22,23,24,25,26]. For instance, our open TLIF technique involved bilateral facetectomy, accurate discectomy, anterior column lengthening, and posterior compression through pedicle screws. These factors could enhance segmental mobility and contribute to the notable increase in SL. In contrast, many studies involving expandable cages used MIS-TLIF, where bilateral facetectomy is often not performed, limiting the capacity to mobilize the spinal segment and apply compressive forces.

The findings of this study have relevant clinical implications for spine surgeons performing open TLIF procedures with expandable cages. The significant and sustained improvements in segmental lordosis, disc height, and foraminal height suggest that these devices can be effective tools for achieving local radiographic correction, particularly when combined with techniques that allow for greater segmental mobilization, such as bilateral facetectomy. Surgeons should be aware that while expandable cages offer advantages in terms of insertion, disc space restoration, and reduced endplate disruption, additional strategies, such as choosing appropriate fusion levels and applying adequate posterior compression, may be necessary to optimize global alignment.

### Strengths and Limitations of the Study

The strengths of this study are that the cases were performed at a single institution by orthopaedic surgeons using three different expandable cage designs. Thus, the study is a real-world evaluation of the performance of expandable cages. However, the study has five limitations. First, the study is retrospective and lacks a control group for comparison. Second, the number of patients who were available for radiographic outcomes measurement at final follow-up was relatively small (71). Third, unavailability of follow-up CT scans for all patients and reliance on X-rays limited the accuracy of subsidence evaluation and precluded determination of fusion rate/outcome. Fourth, only highly experienced orthopaedic surgeons performed the cases. Fifth, many other metrics were not obtained, such as fusion outcome/rate, incidence of complications (for example, cage malposition, cage shrinkage, and screw malposition), and clinical outcomes (for example, PROMS). These limitations should be considered when designing future studies aimed at obtaining definitive determination of the performance of expandable cages in lumbar spine fusion.

## 5. Conclusions

The present results show that with open standard TLIF, use of expandable cages allowed for very good radiological outcomes at 2-year follow-up, notably increase of segmental lordosis and restoration of anterior disc, posterior disc, and foraminal heights. However, it is worth underscoring the fact that many factors play a role in achieving these results, such as appropriate preparation of endplates, posterior facet osteotomy, and posterior compression prior to final fixation.

## Figures and Tables

**Figure 1 bioengineering-12-01169-f001:**
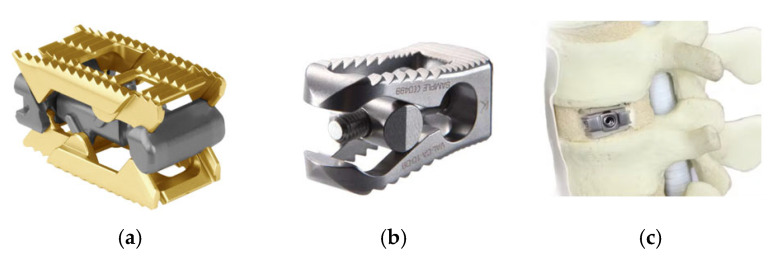
(**a**) Cage Rise (Globus Medical): titanium cage, vertical expansion up to 7 mm. (**b**) Cage L-Varlock (Kisco International): titanium cage, insertion height 8–21 mm, width 10 or 13 mm, lordotic correction up to 24.5°. (**c**) Cage Uplifter (fule Science & Technology Development and Invibio): hybrid PEEK-titanium cage, expands vertically 3–5 mm.

**Figure 2 bioengineering-12-01169-f002:**
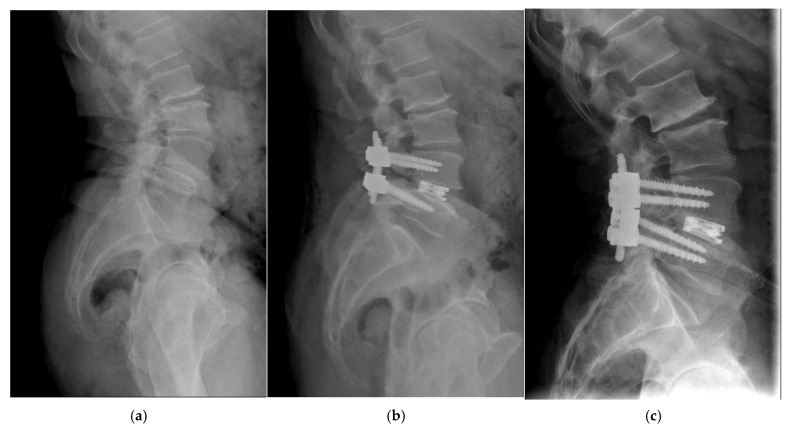
(**a**) Preoperative standing lateral X-ray; (**b**) 1-year postoperative X-ray; (**c**) 2-year postoperative X-ray. The expandable cage used in this case was the L-Varlock.

**Figure 3 bioengineering-12-01169-f003:**
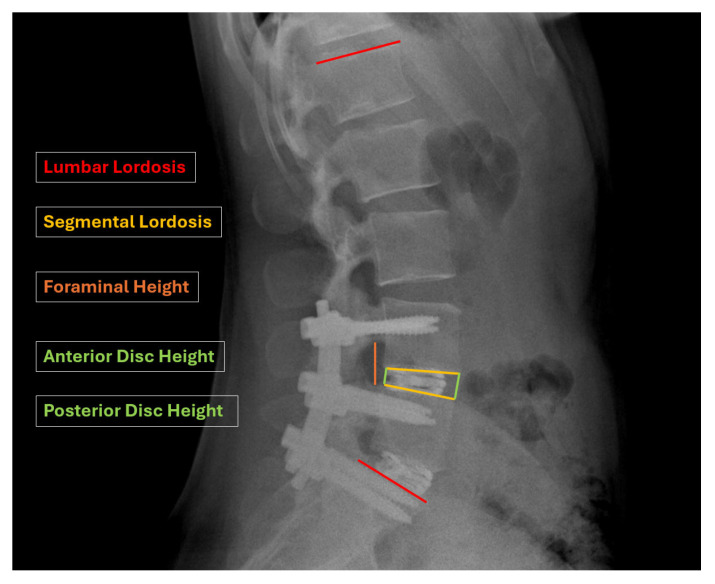
Schematic representation of radiographic measurements, including segmental lordosis (SL), lumbar lordosis (LL) anterior disc height (ADH), posterior disc height (PDH) and foraminal height (FH).

**Figure 4 bioengineering-12-01169-f004:**
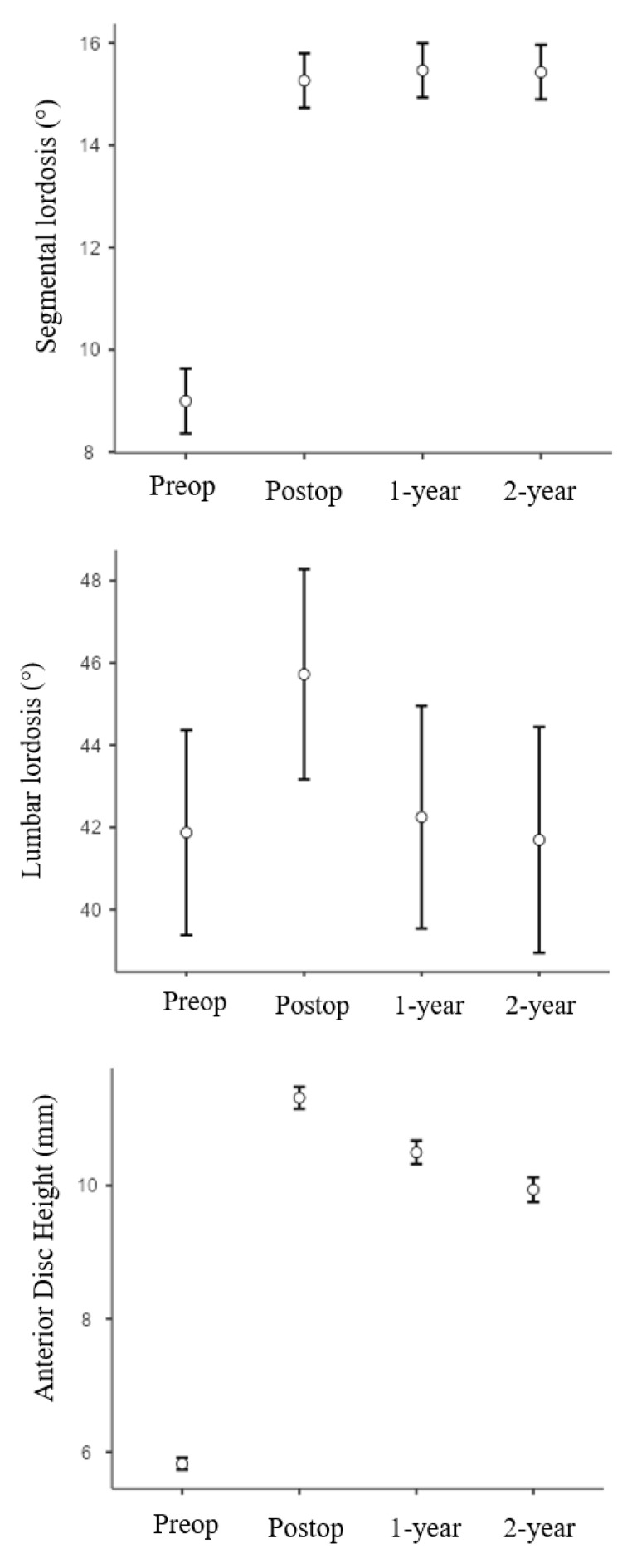
Graphs showing segmental lordosis, lumbar lordosis, anterior disc height, posterior disc height and foraminal height for all patients from preoperative through 24 months follow up (mean ± SD).

**Table 1 bioengineering-12-01169-t001:** Inclusion and exclusion criteria.

Inclusion Criteria	Exclusion Criteria
Degenerative disease from 1 to 3 continuous level(s) between L1 and S1; Age ≥ 18 years; 1–3 levels circumferential lumbar arthrodesis with expandable interbody cages; Minimum radiographic follow-up of 2 years; Failed conservative treatment (minimum 6 months)	Age < 18 years Prior fusion or surgery Trauma Tumor Surgical operation using cages different from expandable cages

**Table 2 bioengineering-12-01169-t002:** Demographic characteristics of the patient population.

Number of Patients	71
Age at the time of surgery (y)	56 (13)
BMI (Kg/m^2^)	25.5 (4.0)
Patients underwent 1-level circumferential arthrodesis	29
L2-L3	1
L3-L4	2
L4-L5	8
L5-S1	18
Patients underwent 2-level circumferential arthrodesis	26
L3-L5	5
L4-S1	21
Patients underwent 3-level circumferential arthrodesis	16
L2-L5	4
L3-S1	12

**Table 3 bioengineering-12-01169-t003:** Radiographic outcomes based on postoperative time interval.

	Preoperative	Postoperative			*p*-Value (Preoperative vs. Immediate Postop)	*p*-Value (Immediate Postop vs. 1 Year FU vs. 2 Years FU)
		**Immediate Postoperative**	**1 Year FU**	**2 Years FU**		
Segmental lordosis (°)	9.0 ± 3.6	15.3 ± 3.0	15.5 ± 3.0	15.4 ± 3.0	*p* < 0.001	*p* = 0.355
Lumbar lordosis (°)	41.9 ± 10.5	45.7 ± 10.8	42.3 ± 11.4	41.7 ± 11.6	*p* = 0.01	*p* = 0.041
Anterior Disc Height (mm)	5.8 ± 0.5	11.3 ± 0.9	10.5 ± 1.0	9.9 ± 1.0	*p* < 0.001	*p* < 0.001
Posterior Disc Height (mm)	3.4 ± 0.6	9.3 ± 0.9	8.6 ± 0.9	8.0 ± 0.9	*p* < 0.001	*p* < 0.001
Foraminal Height (cm)	1.0 ± 0.1	2.3 ± 0.2	2.1 ± 0.2	2.0 ± 0.2	*p* < 0.001	*p* < 0.001

## Data Availability

The original contributions presented in the study are included in the article/Appendix A, further inquiries can be directed to the corresponding author.

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
