# Peer review of "Expandable Interbody Cages in 1–3 Level Circumferential Lumbar Arthrodesis with 2-Year Follow up: A Retrospective Study"

_bioengineering, 2025, doi:10.3390/bioengineering12111169_

Round 1

Reviewer 1 Report

Comments and Suggestions for Authors

PLEASE SEE THE UPLOADED FILE, "REVIEW-REPORT-ON-BIOENGINEERING-MANUSCRIPT-ID-3906608."

Comments on the Quality of English Language

PLEASE SEE THE UPLOADED FILE, "REVIEW-REPORT-ON-BIOENGINEERING-MANUSCRIPT-ID-3906608."

Author Response

Dear Editor,

We would like to sincerely thank you and the reviewers for the suggestions regarding our manuscript entitled “Expandable interbody cages in 1–3 level circumferential lumbar arthrodesis with 2-year follow-up: a retrospective study.”
We have  revised the manuscript accordingly to improve its clarity, accuracy, and overall quality. We hope these revisions improve our work.

Thank you for your kind consideration.

Sincerely,
Bruna Maccaferri
on behalf of all co-authors

Reviewer 2 Report

Comments and Suggestions for Authors
  1. Without a concurrent static-cage or MIS cohort, it is impossible to attribute the observed radiographic changes to the “expandable” feature itself. Provide at least historical controls matched for levels, age, and pre-operative SL/LL, or perform multivariable regression adjusting for cage position, facet resection grade, and compression manoeuvres.
  2. Bridging bone and fusion rate were not assessed (no CT scans). Subsidence ≠ non-union, but non-union may explain late LL loss. Provide either: (a) subset CT at 12-24 months (n≈20) with Lenke fusion grading, or (b) fine-cut CT criteria for subsidence vs. lucency; revise discussion accordingly.
  3. No power calculation. Primary endpoint (ΔSL) effect size 6.3°; provide post-hoc power and 95 % CI for this difference.
  4. Multiple paired t-tests across time points inflate type-I error; use repeated-measures ANOVA with Bonferroni or Dunnett correction, and report effect size (Cohen’s d).
  5. Pelvic incidence (PI), pelvic tilt (PT), sacral slope (SS), PI-LL mismatch are missing. These influence long-term LL and adjacent-segment degeneration. Provide full spinopelvic measurements at each time point and correlate with late LL decay.

Author Response

(The authors gave the same response as above.)

Round 2

Reviewer 1 Report

Comments and Suggestions for Authors

PLEASE SEE THE UPLOADED FILE.

Author Response

Comment 1: Line 92 "Correct to read, “Figure 1. (a) Cage Rise (Globus Medical): titanium …….………”

Answer 1: I revised the manuscript as requested to the reviewer.

Comment 2: Line 206 "Correct to read, “Table 2. Demographic……..”

Answer 2: I revised the manuscript as requested to the reviewer.

Comments 3: Table 2  In column 1, correct, “Age………..(SD)” to read, “Age at……surgery (y)"

In column 1, correct, “Average BMI (SD)” to read, “BMI (kg/m2)”

In column 2, correct “22,5 (4,0)” to read, “25.5 (4.0)”

Answer 3: Thank you for your careful review and for pointing out these details.
The corrections have been made as suggested.

Comment 4: Line 208 Revise the title of the sub-section 3.2 to be, “Radiographic results"

Answer 4: I revised the manuscript as requested to the reviewer.

Comment 5: Sub-section 3.2  Make the first sentence in this sub-section to be, “All the results are summarized in Table 3 and Figure 2.”

Answer 5: I revised the manuscript as requested to the reviewer.

Comment 6: Page 12 Starting with Ref. [19], all subsequent references up to Ref. [26] (line 363-line 379) are out of order in terms of the position at which they were first cited in the text. Please correct this issue.

Answer 6: We thank the reviewer for noticing this issue. The references from [19] to [26] have been carefully checked and reordered to match the sequence in which they are first cited in the text. The numbering in both the text and the reference list has been corrected accordingly.

Reviewer 2 Report

Comments and Suggestions for Authors

I acknowledge the author's revisions and agree to the publication of the article.

Author Response

Dear Reviewer,

Thank you very much for your feedback and for acknowledging the revisions. I appreciate your time and consideration throughout the review process.

Best regards,

Marco Fava